



# The 2019 Raikoke eruption as a testbed for rapid assessment of volcanic atmospheric impacts by the Volcano Response group

Jean-Paul Vernier[1,2], Thomas J. Aubry[4], Claudia Timmreck[3], Anja Schmidt[5,5b,5c], Lieven Clarisse[6], Fred Prata[7], Nicolas Theys[8], Andy T. Prata[9,*], Graham Mann[10], Hyundeok Choi[11], Simon Carn[12], Richard Rigby[13], Susan C. Loughlin[14] and John A. Stevenson[14]

(1) National Institute of Aerospace, Hampton, VA 23692

(2) NASA Langley Research Center, Hampton, VA 23692

(3) Max-Planck-Institut für Meteorologie, Hamburg, Germany

(4) Department of Earth and Environmental Sciences, University of Exeter, Penryn, UK

(5) Institute of Atmospheric Physics (IPA), German Aerospace Center (DLR), Oberpfaffenhofen, German

(5b) Meteorological Institute, Ludwig Maximilian University of Munich, Munich, Germany

(5c) Department of Chemistry, University of Cambridge, Cambridge, United Kingdom

(6) Université libre de Bruxelles (ULB), Service de Chimie Quantique et Photophysique, Atmospheric Spectroscopy, Brussels, Belgium

(7) AIRES Pty Ltd, Mt Eliza, Victoria, Australia

(8) Royal Belgian Institute for Space Aeronomy (BIRA-IASB), Brussels, Belgium

(9) Sub-Department of Atmospheric, Oceanic and Planetary Physics, University of Oxford, Oxford OX1 3PU, UK.

*now at School of Earth, Atmosphere and Environment, Monash University, Clayton, Victoria 3800, Australia.

(10)  University of Leeds, Leeds, UK.

(11) Science Applications International Corporation, Inc. (SAIC) at NOAA/NWS/NCEP/Environmental Modeling Cent

(12) Michigan Tech, USA

(13) Centre for Environmental Modelling and Computation, School of Earth and Environment, University of Leeds, UK

(14) British Geological Survey, Edinburgh, UK

*Correspondence to: Jean-Paul Vernier (jeanpaul.vernier@gmail.com)*



**Abstract.** The 21st June 2019 Raikoke eruption (48°N,153°E) generated one of the largest amounts of sulfur emission to the stratosphere since the 1991 Mt Pinatubo eruption. Satellite measurements indicate a consensus best estimate of 1.5 Tg for the sulfur dioxide ($SO_2$) injected at an altitude of around 14-15 km. The peak northern hemisphere mean 525nm Stratospheric Aerosol Optical Depth (SAOD) increased to 0.025, a factor of three higher than background levels. The Volcano Response (VolRes) initiative provided a platform for the community to share information about this eruption, which significantly enhanced coordination efforts in the days after the eruption. A multi-platform satellite observation sub-group formed to prepare an initial report to present eruption parameters including $SO_2$ emissions and their vertical distribution for the modelling community. It allowed to make the first estimate of what would be the peak in SAOD one week after the eruption using a simple volcanic aerosol model. In this retrospective analysis, we show that revised volcanic $SO_2$ injection profiles yield a higher peak injection of the $SO_2$ mass. This highlights difficulties in accurately representing the vertical distribution for moderate $SO_2$ explosive eruptions in the lowermost stratosphere due to limited vertical sensitivity of current satellite sensors (+/- 2 km accuracy) and low horizontal resolution of lidar observations. We also show that the $SO_2$ lifetime initially assumed in the simple aerosol model was overestimated by 66%, pointing to challenges for simple models to capture how the life cycle of volcanic gases and aerosols depends on the $SO_2$ injection magnitude, latitude and height. Using revised injection profile, modelling results indicate a peak northern hemisphere monthly mean SAOD at 525nm of 0.024, in excellent agreement with observations, associated with a global monthly mean radiative forcing of -0.17 W/m$^2$ resulting in an annual global mean surface temperature anomalies of -0.028 K. Given the relatively small magnitude of the forcing, it is unlikely that the surface response can be dissociated from surface temperature variability.

**1. Introduction.**

After 95 years of dormancy, the Raikoke volcano in the Kuril Islands (North-West Pacific; 48.292°N, 153.25°E) began a series of explosions at 18UTC on 21 June 2019 lasting around 24 hours. Raikoke forms a small uninhabited Island of 2 km x 2.5 km which belongs to the Russian federation, 16 km from Matua Island in the Sea of Okhotsk. Its name originates from the ancient Japanese Ainu language and translate to "hellmouth" referring to past volcanic eruptions. The first eruption reports of Raikoke originated from the mid-18th century but it was during the 1788 eruption that one third of the Island was destroyed (Gorshkov, 1970). The last known eruption was reported in February 1924. Since then, the volcano remained dormant. The volcano is monitored by the Sakhalin Volcanic Eruption Response Team (SVERT) part of the Institute of marine geology and the Kamchatka Volcanic Eruption Response Team (KVERT). During the latest 2019 eruption, the first explosion of a series of 8 was reported by KVERT on 21 June at 17h50 UTC and quickly followed 1h later by a volcanic ash advisory produced by the Tokyo Volcanic Ash Advisory Center (VAAC) responsible to provide ash warnings to the International Civil Aviation Organization (ICAO) across the Pacific Northwest (Sennert, 2019). In addition, KVERT and SVERT issued red warnings for aviation. As a result, nearly 40 flights were re-routed to avoid volcanic ash clouds.

Firstov et al., (2020) analyzed Infrasound Signal (IS) from ground stations in Kamchatka and found a total of 11 explosive episodes (see Fig.1a). The first 8 episodes were followed by a continuous episode (9) which lasted for 3.5 h. Based on IS analysis, episodes are separated into magma fragmentation/ non-stationary processes and vent





outflow (1,2,3,7,9 and 10) of ash-gas into the atmosphere. They were used to derive a minimal eruption tephra
volume of 0.1 $km^3$ allowing to categorize the eruption as Volcanic Explosivity Index (VEI) 4 (Firstov et al., 2020).
Fig1b shows cloud top temperature (11µm) and associated cloud top heights derived from Himawari-8 geostationary
satellite compared with IS data shown in Fig.1a. The eruption started at around 18:00 UTC on 21 June 2019
followed by at least 8 discrete "bursts" (eruptions) and continuous emissions.  A further two discrete pulses occurred
later. The IS analysis coincides very well with the Himawari-8 observations where each IS corresponds to the
release of volcanic cloud into the atmosphere. Muser et al. (2020) used one-dimensional volcanic plume models
(Mastin, 2007; Folch et al., 2016) to invert the mass eruption rate of ash and initialize the ICON-ART (Zängl et al.,
2015) dispersion model to investigate the complex aerosol, dynamical and radiative processes governing the plume
evolution. More simplistic initialization approach with the dispersion model NAME (Beckett et al., 2020) and the
aerosol-chemistry-climate model WACCM (Mills et al., 2016) were performed during the VolRes activities shortly
after the eruption to assess the early dispersion of the plume.
As part of the scientific response to the eruption, the Volcano Response (Volres) initiative triggered an initial
dialogue among the science community. VolRes is an international working group, within the Stratospheric Sulfur
and its Role in Climate (SSiRC) to establish co-operation and community planning, for the next large-magnitude
eruption, aligned also to the NASA initiative for US-based volcano response plan (Carn et al., 2021).  The SSiRC
initiative is itself an activity within the SPARC project of the World Climate Research Program (WCRP). Since its
inception in 2015, VolRes consist of more than 250 scientists worldwide, from a diverse range of both model and
observational experts, aiming to contribute from sharing and discussion of information related to the atmospheric
impacts of volcanoes. Discussion and sharing to the mailing list is maintained through an archive and Wiki page,
structured by eruption since 2018 (https://wiki.earthdata.nasa.gov/display/volres[2]).
The discussions on the VolRes forum have mostly been focused towards: i) establishing initial estimates of the
emitted $SO_2$ and ash, and injection heights estimates from multiple satellite observation platforms; ii) the expected
impacts on stratospheric aerosol loadings; iii) factors to consider in modelling the aerosol cloud, towards then
projecting radiative and climate effects; and iv) common related findings after other similar eruptions. Several cross-
institutional co-operations resulted from the VolRes activity, which also motivated the Raikoke ACP/AMT/GMD
inter-journal special issue "Satellite observations, in situ measurements and model simulations of the 2019 Raikoke
eruption ". The Raikoke special issue includes a series of publications (Muser et al., 2020; Kloss et al., 2021;
Vaughan et al., 2021; de Leeuw et al., 2021; Horváth et al., 2021a,b; Gorkavyi et al., 2021; Inness et al., 2022;
Mingari et al., 2022; Osborne et al., 2022; Bruckert et al., 2022; Capponi et al., 2022; Cai et al., 2022; Harvey et al.,
2022; Knepp et al., 2022; Prata et al., 2022; Petracca et al., 2022) focusing on the atmospheric impacts of this
eruption using satellite Low Earth Orbiting/Geostationary nadir and limb observations from UV-Visible to far IR,
model simulations, airborne measurements and ground-based lidar observations.
The goals of this paper is to:
•    Describe the activities undertaken by the Volcano Response group (VolRes,

https://wiki.earthdata.nasa.gov/display/volres/Volcano+Response) at the time of the 2019 Raikoke eruption. A





chronology of these activities is provided in Table 2.
• Give an overview of the early estimates of the mass of SO$_2$ emitted as well as the associated radiative forcing
and temperature response inferred quickly after the eruption.
• Discuss how revised estimates of SO$_2$ mass and plume heights as well as radiative forcing estimates differ from
the rapid assessment made a week after the eruption.
• Summarize the findings of the Raikoke special issue and highlight the remaining science questions as well as
the challenges associated with rapid response to volcanic eruptions in the context of atmospheric impacts.
**2. Satellite Datasets**
**HIMAWARI-8**
Himwari-8 is a spacecraft developed and operated by the Japanese Meteorological Organization (JAXA). The
primary instrument aboard Himawari 8 is the Advanced Himawari Imager (AHI), a 16 multi-channel spectral
imager to capture visible light and infrared images of the Asia-Pacific region at 500m horizontal resolution and
every 10 minutes. AHI is used to derived the cloud-top temperature and associated cloud top height associated with
the Raikoke eruption.
**TROPOMI**
The TROPOspheric Monitoring Instrument (TROPOMI), on board the Sentinel-5 Precursor satellite provides
atmospheric composition measurements (Veefkind et al., 2012) at high spatial resolution of 3.5 x 5.5 km².
TROPOMI is a hyperspectral sounder with different spectral bands from the ultraviolet (UV) to the short-wave
infrared. TROPOMI provides nearly global coverage in one day at 1.30 pm local time. For a rapid assessment of the
total emitted SO$_2$ mass, the operational SO$_2$ product (Theys et al., 2017) was used. A refined analysis was then
performed with the scientific SO$_2$ layer height and vertical column joint retrieval of Theys et al.(2022)
**IASI**
The Infrared Atmospheric Sounding Interferometer (IASI) is the high spectral resolution infrared sounder onboard
the operational Metop A-B-C platforms. With a morning and evening overpass (around 9:30 AM and PM),
combined with a large swath, the instrument samples the entire globe twice a day. Its footprint is a 12km diameter
circle at nadir viewing angles, gradually increasing to a 20 km x 39 km ellipse at the far end of its swath. The SO$_2$
product that was used for rapid assessment is the one detailed in Clarisse et al. (2014). The retrieval algorithm
consists of two steps. First a so-called Z function that is estimated for each observed spectrum, using a set of
derivatives (Jacobians) with respect to the SO$_2$ partial columns at varying altitudes. The altitude at which Z function
reaches is maximum is the retrieved SO$_2$ height.  In a second step, the estimated SO$_2$ height is used to constrain the
IASI SO$_2$ column retrieval.  Note that the entire retrieval uses the 7.3 µm absorption band of SO$_2$, which is less
affected by ash than the 8.6 µm band. While the altitude algorithm has a general accuracy better than 2 km, it is
known to underestimate the SO$_2$ altitude for high SO$_2$ columns. For the refined analysis discussed below, a new
experimental product was used that deals better with saturation issues.



**Aqua/AIRS**
The atmospheric Infrared Radiation Sounder (AIRS) instrument is on board the NASA polar-orbiting Aqua satellite
at an altitude of about 705 km above the Earth surface with an Equatorial crossing time at 1.30am/pm local time
(Chahine et al., 2005; Prata & Bernardo, 2007). AIRS provides nearly continuous measurement coverage during
14.5 orbits per day and a 95% global daily coverage with a swath of 1650 km and special resolution of 13.5 km x
13.5 km at nadir (Tournigand et al., 2020). We use the version 7.0 AIRS level 2 Support Retrieval product, and the
results are averaged into 1° x 1° grid cells in this analysis. The brightness temperature difference (less than -6 K) is
used as a proxy of $SO_2$ released from volcanoes.
**CALIPSO/CALIOP**
The Cloud-Aerosol Lidar with Orthogonal Polarization (CALIOP), on board the Cloud-Aerosol Lidar and Infrared
Pathfinder Satellite Observations (CALIPSO) platform, has been providing aerosol vertical profile measurements of
the Earth's atmosphere on a global scale since June 2006 (Winker et al., 2010). We use the version 4.21 CALIOP
level 2 Aerosol layer and Cloud layer products and only quality screened samples are used in the analysis. Aerosol
layers with Cloud Aerosol Discrimination (CAD) score less than -100 or greater than -20 are rejected to avoid low
confidence in cloud-air discrimination. Aerosol layers with the extinction Quality Control (QC) flag that are not
equal to 0, 1, 16, and 18 are rejected to remove low confidence extinction retrievals, and aerosol extinction samples
with the extinction uncertainty equal to 99.99 km$^{-1}$ and all samples at lower altitudes in the profile are rejected to
remove unreliable extinctions (Winker et al., 2013).
Firstov et al., (2020) analyzed Infrasound Signal (IS) from ground stations in Kamchatka and found a total of 11
explosive episodes (see Fig.1a). The first 8 episodes were followed by a continuous episode (9) which lasted for 3.5
h. Based on IS analysis, episodes are separated into magma fragmentation/ non-stationary processes and vent
outflow (1,2,3,7,9 and 10) of ash-gas into the atmosphere. They were used to derive a minimal eruption tephra
volume of 0.1 km$^3$ allowing to categorize the eruption as Volcanic Explosivity Index (VEI) 4 (Firstov et al., 2020).
Fig1b shows cloud top temperature (11μm) and associated cloud top heights derived from Himawari-8 geostationary
satellite compared with IS data shown in Fig.1a. The eruption started at around 18:00 UTC on 21 June 2019
followed by at least 8 discrete "bursts" (eruptions) and continuous emissions.  A further two discrete pulses occurred
later. The IS analysis coincides very well with the Himawari-8 observations where each IS corresponds to the
release of volcanic cloud into the atmosphere. Muser et al. (2020) used one-dimensional volcanic plume models
(Mastin, 2007; Folch et al., 2016) to invert the mass eruption rate of ash and initialize the ICON-ART dispersion
model to investigate the complex aerosol, dynamical and radiative processes governing the plume evolution. More
simplistic initialization approach with the dispersion model NAME and the aerosol-chemistry-climate model
WACCM were performed during the VolRes activities shortly after the eruption to assess the early dispersion of the
plume.

**4. Early reports of injection parameters one week after the eruption**





One of the main activities of a satellite sub-group formed within the framework of VolRes was to derive eruption
parameters characterizing $SO_2$ emissions (e.g. mass, bulk height, injection profiles) so that modelers would run
numerical simulations to understand the potential hazards and climate impacts of this eruption. The basic approach
to estimate the total mass of $SO_2$ is similar for each satellite-based sensor. First, the process involves retrieving the
Vertical Column Density (VDC, measured in molecules $cm^{-2}$ or $g\ m^{-2}$ or Dobson units) in each pixel affected by
$SO_2$, followed by multiplying by the area of the pixels and integrating all the pixels to calculate the total $SO_2$
loadings. However, there are limitations to this method. Indeed, narrow swath width sensors, timing of the polar
orbit and, in the case of the geostationary sensors, extreme viewing geometry (high satellite zenith angles) and
movement out of the field of view will introduce errors (likely underestimations) of the total mass. There are also
many assumptions used by the various algorithms that if not valid will introduce errors, as will discussed hereunder.
When the Vertical Column Densities (VCDs) are large (>500 DU) most algorithms have difficulty estimating the
VCD correctly (Hyman and Pavolonis, 2020; Prata et al. 2021). Figure 2 shows the time evolution of the total $SO_2$
mass during and after the Raikoke eruption from multiple sensors. The measurements discussed here all assume $SO_2$
in the UTLS (7–12 km). The $SO_2$ retrieved from Himawari-8 peaks near 1.5 Tg nearly 48h after the beginning of the
eruption and follow similar temporal evolution than the one derived from LEO. Given the likelihood that most
satellites underestimated the $SO_2$ mass, we chose at that time the maximum value from Himawari and the upper
limits of the other sensors yielding a 1.5+/-0.2 Tg estimation.  IASI, TROPOMI and CALIPSO data suggested that
$SO_2$ was injected within a large altitude range from the ground up to well in the stratosphere (at least 15 km). In
addition to a total mass of $SO_2$ (of 1.5 Tg), the VolRes team also issued a provisional vertical distribution of the
emitted $SO_2$ mass that could be used by dispersion and climate modelers. To do so, IASI $SO_2$ height measurements
on the $22^{nd}$ June 2019 were used. The mass-altitude indicated that most $SO_2$ was released between 8-12 km with a
secondary peak around 14-15 km. Scaled to the proposed 1.5 Tg, the distribution is shown in Figure 3 and is referred
to as the 'VolRes profile' (blue line; also see Table 1).  For TROPOMI, and other LEOs, the plume can be partly
covered by a given orbit but using the multiple orbits of one day and the fact that they generally overlap most of the
plume is covered. To avoid double counting, the data of one full day are usually averaged on a regular latitude-
longitude grid, before the actual emitted $SO_2$ mass is calculated.  An important source of error is the vertical
distribution of $SO_2$. In Fig.2, the retrieved $SO_2$ mass from TROPOMI was calculated by assuming a bulk plume
height of 15 km (all plume heights given above sea level unless specified). This assumption can introduce errors
(underestimation) in particular for clear-sky scenes and if the $SO_2$ is in the (lower) troposphere, typically below
7km, see e.g., Fig 1 of Theys et al. (2013). TROPOMI has less limitations in retrieving very large $SO_2$ columns
(>500 DU) because in that case the spectral range used (360-390nm) is weakly affected by saturation due to non-
linear $SO_2$ absorption (Bobrowski et al., 2010). The main problem is the presence of aerosols which are not
explicitly treated in the retrievals (Theys et al., 2017). For ash, the photons cannot penetrate deep in the volcanic
cloud (only the cloud top layer is sensed) and this leads to a strong underestimation of the mass of $SO_2$ (by a factor
of 5 or so).
**5. Revision and improvements of injection parameters.**





While the accuracy of the IASI SO$_2$ height retrievals is typically better than 2km, it became clear however that the
VolRes profile was peaking too low in the atmosphere (e.g., de Leeuw et al., 2021). The main reason for this is related
to the SO$_2$ Jacobeans used in the retrieval. These are precalculated for relatively low SO$_2$ VCDs and are not directly
applicable to saturated plumes, as encountered during the Raikoke eruption. Refinement of the IASI algorithm to
better account for this dependence on the SO$_2$ loadings has led to SO$_2$ injection profile with a maximum SO$_2$ peaking
at ~14-15 km (see Figure 3) and a slightly lower total mass of ~1.3 Tg SO$_2$ (even though total mass estimates for the
days after reach again 1.5 Tg and higher).
As an alternative to IASI, ultraviolet observations from the TROPOMI nadir sensor have been used to estimate the
SO$_2$ injection profile (Table 1). Conceptually, the retrieval algorithm is like the IASI scheme. It relies on an iterative
approach making use of a SO$_2$ optical depth look-up-table, where both SO$_2$ height and vertical column are retrieved
jointly (Theys et al., 2021). The accuracy of the retrieved SO$_2$ heights is of 1-2 km, except when coincident with fresh
and optically thick ash plumes for which the estimated heights can be strongly biased low. Because of this, the first
reliable profile from TROPOMI which covers the full plume, is for the 24 June 2019. The maximum SO$_2$ height is
found at ~11-12 km (Figure 3) and the total mass derived is of ~1.2 Tg SO$_2$. However, the total mass is likely
underestimated because only the pixels with confident SO$_2$ height retrievals are considered (typically for SO$_2$ columns
> 5DU). Selected examples of retrieved SO$_2$ heights from the two instruments are illustrated in Figure 4.
Although the estimated SO$_2$ mass from IASI and TROPOMI agree well, the estimated SO$_2$ profiles show rather
inconsistent results with a discrepancy of about 3km for the SO$_2$ bulk height.  It should be emphasized that SO$_2$ height
retrieval from nadir sensors is challenging in general but for Raikoke in particular. The retrievals and their
interpretation might also suffer from different aspects. For instance, the UTLS was characterized by isothermal
temperature profiles, which can lead to errors on the IASI height estimates. In addition, the measurement sensitivity
is different in the ultraviolet than in the thermal infrared and depends on the way the photons interact with the volcanic
cloud (and the constituents other than SO$_2$). In this respect, the retrieved SO$_2$ heights must be considered as effective
heights. Moreover, few CALIOP observations were available (see Section 6) for evaluating the results for the early
stage of the eruption.
Despite these challenges, our injection profiles estimates are not in contradiction with results found in the literature:
• Kloss et al. (2021) reported a 14 km altitude plume height based on an early OMPS aerosol extinction profile,
on 22 June 2019.

• Muser et al. (2020) derived typical altitudes of 8-14 km from MODIS and VIIRS cloud top height retrievals.
• By slightly adapting (assuming higher injection heights) the VolRes profile, de Leeuw et al. (2021) found
the best match between modeled and TROPOMI SO$_2$ columns for an injection profile with most of SO$_2$
between 11 and 14 km.

• Hedelt et al. (2019) reported SO$_2$ heights similar to the TROPOMI results shown here, i.e., with the bulk
height below 13km.





- •   $SO_2$ height retrievals from the Cross-track Infrared Sounder (CrIS) instrument (Hyman & Pavolonis, 2020)
- are consistent with plume heights as high as 14-17 km in the plume center, but also show that most of the
- $SO_2$ mass was emitted under 13 km.
- •   Geometric estimation of Raikoke ash column height suggests injection mainly between 5 and 14 km and an
- overshooting cloud up to 17 km (Horváth et al., 2021b).
- •   MLS data for 23-27 June indicates $SO_2$ plumes at 11 to 18 km with maximum columns observed around 14
- 250         km (Gorkavyi et al., 2021).
- •   Using a Langragian transport model combined with TROPOMI and AIRS, Cai et al. (2022) reconstruct an
- emission profile with a peak at 11 km with a large spread from 6 to 14 km.
- •   Prata et al. (2022) found ash clouds at a maximum height of 14.2 km (median height of $10.7 \pm 1.2$ km) during
- the main explosive phase.

**6. New plume injection analysis derived from CALIPSO and AIRS**
CALIPSO observations were made publicly available within 24-48 h after the beginning of the eruption allowing
accurate early estimates of the height of downwind plume sections. However, due to the narrow swath of the lidar (a
few hundred meters) and consequently low horizontal resolution, they may not completely represent the entire
plume vertical distribution. Nevertheless, an overpass of the CALIPSO lidar across the plume on 22 June 2019 at
2.15 am, ~600 km east from the volcano within an $SO_2$ cloud observed by OMPS show volcanic layers between 9-
13.5 km (Prata et al., 2021). A second overpass the next day depicts another volcanic layer between 15-16 km.
Those observations were used to validate $SO_2$ emission profiles provided to the community a week after the
eruption. Here, we give a more comprehensive analysis of the plume injection height using a combination of quasi-
collocated (less than 1h apart) $SO_2$ observations from AIRS and detected volcanic layers from CALIOP during the
first two weeks after the eruption. The brightness temperature difference (1361.44-1433.06 $cm^{-1}$) is used as a proxy
of $SO_2$ released from volcanoes to identify CALIOP data within the $SO_2$ plume.
We combined $SO_2$ information from AIRS quasi-collocated observations from CALIOP to further investigate plume
injection heights after the Raikoke eruption assuming that $SO_2$ and volcanic aerosols remained collocated in space
and time during the first 10 days after the eruption. Figure 5 shows a map of $SO_2$ derived from AIRS together with
CALIOP orbit tracks (red). The corresponding cloud and aerosol level 2 V4.2 products are plotted along with BTD
extracted along the orbit. All corresponding layers (clouds and aerosols) associated with negative BTD (<6 K),
indicating the presence of $SO_2$ in the atmospheric column, have been further analyzed to distinguish the volcanic
plume. The distinction is based on the diagram of depolarization and color ratio shown in panel d. Figure 5 shows
that CALIOP intersected the plume along two orbit tracks on 25 June. The first being along the 17h53 UTC orbit
near 60°N and at two occasions between 55°N-65°N along the second orbit near 14h36 UTC. The first intersection
shows the plume near 9-11 km with weak particulate DePolarization Ratio (DPR) (DPR < 0.2) and particulate
CoLor Ratio (CLR) near 0.5. DPR values suggest a mixture of ash and sulfate aerosols. However, the second
intersection of the plume shows higher DPR near 0.3 and the same CLR than the first indicating a higher fraction of
ash particles resulting in increased DPR values. During those observations, two distinct plumes are visible between



the northern intersection near 11-13 km (green color on diagrams) and a piece at higher altitude (13-15 km) further
south (<60°N). We visually inspected all CALIOP observations (day and night) between 06/22 and 07/06 following
the same approach and used plume identification criterion when DPR < 0.4 and CLR < 0.7 and altitude > 5 km to
remove tropospheric aerosols and ice clouds. Because of the enhanced noise of the daytime observations, we chose
to focus this analysis on nighttime data only. Figure 6 shows the daily observations of the Raikoke plume since the
eruption and during the following two weeks. We note that the plume was observed by CALIOP from 8 km to 17
km. The cumulative Probability Density Function (pdf) suggests two main peaks, one near 10-11km km and another
smoother peak near 13-15 km. The overall aerosol vertical distribution is consistent with the distribution of $SO_2$
profiles derived with different approaches and instruments just after the eruption (Fig.3). However, the pdf does not
suggest a pronounced peak at a given altitude but rather a flatter distribution as opposed to what is shown in Figure
3. The pdf does not account for or is not weighted by the aerosol loading which may explain why we do not see a
pronounced peak as for the $SO_2$ profiles derived from IASI and TROPOMI. In addition, $SO_2$ and volcanic aerosol
layers are assumed to be collocated but it may not always be the case.
**7. Rapid projections of the aerosol forcing and the global mean surface temperature response.**
In the previous sections, we discussed in detail the methods used to derive injection parameters ($SO_2$ total mass,
plume heights and $SO_2$ distribution) which served as input to estimate the radiative and surface temperature
responses from the eruption in this section. Key metrics characterizing the climate effects of volcanic eruptions are
the peak global mean mid-visible SAOD, the global mean net radiative forcing and the global mean surface
temperature change. One motivation of the VolRes initiative is to provide an estimated magnitude for each of these
metrics. In the case of a large-magnitude eruption, these initial indicators of the scale of the climate response would
then help to determine whether resources should be directed towards additional measurement campaign and the
forcing datasets enable the community to run seasonal and decadal forecasts (Müller and Smith, 2018).
The first estimates of the injected $SO_2$ mass and height became available 24-48 hours after the 2019 Raikoke
eruption, followed one week later by an estimate of global mean peak SAOD (7.1), radiative forcing (7.2) and
surface temperature (7. 3). This section discusses: i) how these estimates were made; ii) how they compared to
observations; and iii) ongoing improvements to the protocol for rapid projection of volcanic forcing and climate
impact.
**7.1 Model simulations of aerosol optical properties**
We first made projections for SAOD on 25 June 2019 using EVA_H (Aubry et al., 2020), a simple volcanic aerosol
model based on inputs of the mass of volcanic $SO_2$ injected, its injection height, and the latitude of an eruption. The
first estimates made following Raikoke used a range of injection heights between 10-20 km, and a range of the mass
of $SO_2$ of 1-2 Tg of $SO_2$, on the basis of first estimates of 14 km and 1.5 Tg of $SO_2$ that initially circulated on the
VolRes mailing list (personal communication from Taha Ghassan and Lieven Clarisse). The corresponding
simulated range in peak Northern Hemisphere (25ºN-90ºN, NH) monthly-mean SAOD at 525nm ($SAOD_{525}$) was
0.015-0.023 (Figure 7). This range was obtained using Monte Carlo methods, i.e. EVA_H was run thousands of



times randomly resampling the range of injection height and mass. The negligible computational cost of simple
models like EVA_H is a key advantage for providing estimate of the volcanic SAOD perturbation and its
uncertainties as soon as measurements of the $SO_2$ mass and its injection height become available. The SAOD
perturbation was projected to be largely confined to 25-90ºN (Figure 8). SAOD perturbations observed in the tropics
and Southern Hemisphere over 2019-2020 (Figure 8) are primarily driven by stratospheric emissions from the
Ulawun 2019 eruptions and the Australian 2019-2020 wildfires (Kloss et al., 2021).
Following the communication of the initial VolRes $SO_2$ profile (Figure 3) through the VolRes mailing list, EVA_H
peak NH monthly-mean $SAOD_{525}$ estimate for Raikoke were revised to an even smaller value of 0.014. Compared to
observations from GloSSAC (v2.1) (Kovilakam et al., 2020), this value was largely underestimated as GloSSAC NH
monthly-mean $SAOD_{525}$ peaks at 0.025 (Figure 7, with GloSSAC in excellent agreement with observational values
from Kloss et al., 2021) using OMPS-limb data. The new IASI June 22 profile presented in Figure 3 results in a
higher peak NH monthly-mean $SAOD_{525}$ of 0.0175, with the higher proportion of stratospheric $SO_2$ in the new
profile more than compensating for the total mass decreasing from 1.5 to 1.29 (average of the two IASI profiles) Tg
of $SO_2$. Although the new $SO_2$ emission profile improves agreement with observations, the estimated $SAOD_{525}$
value is still a substantial underestimate. Furthermore, the characteristic rise and decay timescales of the $SAOD_{525}$
perturbation are also overestimated by EVA_H (Figure 7). These mismatches are caused by the constant timescale
EVA_H uses for $SO_2$ to sulfate aerosol conversion, which is biased towards an 8-month value adequate for the
Pinatubo 1991 eruption (Aubry et al, 2020). If we decrease the value of this timescale by 66% to 2.8 month in
EVA_H, the NH peak SAOD value as well as the characteristic rise and decay timescale of the SAOD perturbation
are in excellent agreement with observations for the 2019 Raikoke eruption (Figure 7). The fact that this model
timescale is independent of the eruption characteristic is an already identified weakness of EVA_H that will be
addressed in future developments (Aubry et al., 2020). This timescale has indeed been shown to depend on the
volcanic $SO_2$ mass (e.g. McKeen et al., 1984; Carn et al, 2016), injection altitude and latitude (e.g. Carn et al, 2016,
Marshall et al. 2019) as well as co-emission of water vapor (Legrande et al., 2016) and volcanic ash (Zhu et al.,

2022).

**7.2 Projection for global mean volcanic forcing**
On the same day that SAOD projections were initially provided, Piers Forster independently suggested via the
VolRes mailing list (Forster, personal communication) that the global annual-mean net radiative forcing would be at
most -0.2 W m$^{-2}$ based on a scaling between the estimated $SO_2$ mass of 1.5 Tg $SO_2$ for 2019 Raikoke and the
estimated 15-20 Tg $SO_2$ for the 1991 Mt. Pinatubo eruption, which resulted in a global annual-mean forcing of -3.2
W/m$^2$ in 1992. This projection was a back-of-the-envelope calculation using simple proportionality arguments and it
did not rely on any SAOD estimates. A monthly global mean peak shortwave forcing with a range from −0.16 to
−0.11W/m$^2$ was derived from SAGE III observations (Kloss et al., 2021). The corresponding annual mean net
forcing is expected to be much smaller because of the difference between the peak monthly NH mean SAOD and its
average value over the first post-eruption year (Figure 7), as well as the fact that longwave stratospheric volcanic



aerosol forcing can offset as much as half of the shortwave forcing (Schmidt et al. 2018). Altogether, the educated
guess made for global annual mean radiative forcing was thus likely overestimated.

### 7.3 Projection of the global mean surface temperature response

Last, as part of the eruption response, one day after the first global annual-mean radiative forcing estimate of 0.2 W
m$^{-2}$ was made, we estimated that the peak global annual-mean surface temperature change would be -0.02 K (Figure
9). We obtained this estimate using FaIR, a simple climate model (Smith et al., 2018). Like EVA_H, FaIR has a
negligible computational cost enabling rapid estimates of global-mean surface temperature change following an
eruption and facilitating uncertainty estimation, although the latter was not done for the 2019 Raikoke eruption. The
model-projected surface temperature response cannot be compared to measurements owing to difficulties in
disentangling such a small forced temperature response from temperature variations related to natural variability.

### 8. Discussions

The Raikoke eruption ended a period without moderate volcanic eruptions in the Northern Hemisphere since Nabro
in 2011 (Bourassa et al., 2013, Fairlie et al., 2014; Sawamura et al., 2012) which injected 1.5-2 Tg of $SO_2$ partially
distributed between the troposphere and stratosphere. Following the Nabro eruption, the role deep convection during
the Summer Asian Monsoon was evoked to explain an apparent ascent of the plume (Bourassa et al., 2013) debated
by others (Fromm et al., 2013, Vernier et al., 2013) based on initial observations of injection heights. The substantial
debate provoked by this eruption clearly demonstrated the complexity of assessing accurately $SO_2$ injection heights
and their partition relative to the tropopause. The VolRes initiative substantially helps fill those gaps by providing a
coordinated structure to derive injection parameters after the Raikoke eruption. Multiple sensors were used to assess
the total $SO_2$ mass and its distribution just one week after the eruption (Fig.3). However, the lack of vertically
resolved $SO_2$ information remains a limitation to accurately assess $SO_2$ plume distribution and the revised estimates
proposed here remain with a 2 km uncertainty regarding the exact position of the plume peak while the initial 1.5 Tg
$SO_2$ mass estimate might be slightly overestimated. Advances in measuring $SO_2$ with lidar observations may fill
those gaps in the future.
The VolRes team provided eruptive parameters within a week after the eruption that strongly helped modelers to
estimate climate response of the Raikoke eruption. The use of simple models like EVA_H and FaIR to project the
climate response to an eruption in almost near real-time is a powerful way to generate first-order estimates of the
perturbations to SAOD, and surface temperatures.  Unlike simple proportionality arguments based on the Pinatubo
1991 eruption, these models can estimate the time (and spatial, for EVA_H) evolution of the response variable, and
they account for complexities such as the dependency of SAOD on the $SO_2$ injection latitude and height. Their
computationally inexpensive nature also enables a comprehensive quantification of uncertainties related to eruption
source parameters, which are often poorly constrained in the days-months following an eruption as highlighted by
this special issue, as well as uncertainties on parameters of these empirical models, such as the $SO_2$-aerosol
conversion timescale in EVA_H (Figure 7).



One limitation of the application of these models following the Raikoke 2019 event is that they were not applied in
concordance, i.e. FaIR was run using an expert guess for the radiative forcing instead of values derived from
EVA_H's SAOD estimates (see section 7.2 and 7.3). Following the Raikoke 2019 VolRes response, we combined
the simple models EVA_H (for aerosol forcing) and FaIR (for surface temperature response). To do so, we apply
simple linear (Schmidt et al., 2018) or exponential (Marshall et al., 2020) relationships to derive the global mean
radiative forcing (FaIR's key input) from the global mean SAOD (one of EVA_H's outputs). EVA_H, SAOD-
radiative forcing scalings, and FaIR were for example applied in concordance to estimate the climate impacts from
the sulfate aerosols of the January 2022 Hunga Tonga-Hunga Ha'apai eruption. These models have been combined
into a single dedicated webtool called Volc2Clim (Schmidt et al., 2023), publicly available at
https://volc2clim.bgs.ac.uk/. Applied to Raikoke 2019 using the new injection profile (Figure 3) and revised $SO_2$ to
sulfate aerosol conversion timescale, the beta version of Volc2Clim projected peak global mean of 0.008, -0.17
$W/m^2$ and -0.028 K for monthly mean SAOD, monthly mean radiative forcing and annual mean temperature
anomaly. In addition to key metrics discussed in this section such as global mean SAOD, radiative forcing and
surface temperature, aerosol optical properties field (dependent on latitude, altitude and wavelength) are outputted
by Volc2Clim for use in climate models that do not have an interactive stratospheric aerosol scheme. With a webtool
for rapid estimation of the global climate response during an eruptive crisis, we hope to support communication
amongst the scientific community (including VolRes), with authorities and with the public, which in turn will help
to mitigate potential consequences arising from the climate effects of an eruption.
Although Volc2Clim offers new perspectives for rapid response and communication following volcanic eruptions,
the simplified nature of the models at its core currently do not allow projections of effects related to co-emission of
species such as water vapor or halogen in volcanic plumes, or PyroCumulonimbus (PyroCbs) plumes. Before and
after the Raikoke eruption, three significant events affected stratospheric aerosols. Indeed, $SO_2$ injected from the
June an August 2019 Ulawun eruptions and smoke from PyroCbs in Canada made the Raikoke eruption even more
challenging to understand. The PyroCbs in Canada produced smoke in the UTLS one week before the eruption, but
the transport patterns of smoke and volcanic aerosols have been distinct (Osborne et al., 2022) and the likelihood for
both plumes to mix is relatively small. The Ulawun eruption injected $SO_2$ which remained relatively confined in the
Southern Hemisphere, but we cannot rule out that both plumes got mixed in the tropics (Kloss et al., 2021).  The
relatively small amount of $SO_2$ injected by Ulawun (< 0.1 Tg) was not considered in the estimates provided in this
paper. Another interesting feature observed after the Raikoke eruption was the formation of a distinct plume which
rose into the stratosphere. The plume formed a vortex circulation which remained coherent for several weeks
(Gorkavyi et al., 2021) rising in the stratosphere of 10 km over the course of 2-3 months. While this plume shared
similar optical properties to smoke, Knepp et al. (2022) concluded that this layer was mostly composed of large
sulfuric acid droplets but did not refute the possible presence of a fine ash component. More recently (Khaykin et al,
2023) found that 24% of the total $SO_2$ mass was contained in the volcanic vortex with a confined anticyclonic
circulation detected by wind doppler lidar from Aeolus. A warm anomaly of 1 K was also evident GPS RO Cosmic
data demonstrating that the heating of the plume was indeed responsible for its internal circulation and maintenance.
Moreover, the properties of the plume observed by CALIOP showed the persistence of ash that likely induced



internal heating in the plume consistent with earlier observations of volcanic clouds after the Kelud and Puyehue-
Cordon eruptions (Jensen et al., 2018; Vernier et al., 2013, 2016). While the presence of fine ash in the Raikoke
could likely explained the maintenance of the vortex as observed after PyroCbs events but with a much faster ascent
rate, the interplay between ash and sulfate and influence on radiative calculations is still not understood (Vernier et
al., 2016; Stenchikov et al., 2021; Zhu et al., 2020). In addition, we cannot fully rule out that remnants of smoke
from the PyroCbs in Canada one week before the eruption could have played a role in the transport of the plume.
The increased lifetime of this plume may have produced a larger climate impact than expected since this effect is not
included in the simple model provided in this paper (Figure 8).
Finally, the recent eruption of Hunga Tonga Hunga Ha'apai demonstrated that sub-marine eruption can inject
significant amount of $H_2O$ in the stratosphere (Milan et al., 2022, Vogel et al., 2022; Sellitto et al., 2022) which is
known to have oppositive cooling climate effects than sulfate aerosol. The water vapor can reduce the lifetime of
$SO_2$ by providing OH radicals and affect aerosol size distribution through condensational growth (Zhu et al., 2022).
Such effects are not included in the simple climate estimates provided here and would limit its applicability in the
case of HTHH if only the climate impacts of sulfate aerosols are considered.

## 435 9. Conclusion

VolRes is an international coordinated initiative to study the atmospheric impacts of volcanic eruptions, now
involving more than 250 researchers worldwide. The 2019 Raikoke eruption triggered significant responses by the
VolRes community through exchanges of information via the mailing list and the preparation of $SO_2$ profile
recommendations for modelers made available a week after the eruption only. Our paper gives a brief overview of
how the community responded to this volcanic eruption, which is documented extensively in the Raikoke special
issue. We then described how early estimates of $SO_2$ emission and height, a fundamental parameter which dictates
the plume lifetime and its impacts, were derived from satellite observations. These estimates were used by VolRes to
calculate SAOD, radiative forcings and surface temperature changes as part of the initial eruption response. We
revisited the initial $SO_2$ injection profiles by addressing saturation effects due to high $SO_2$ column density to
improve plume injection heights. We highlight remaining challenges in accurately representing the vertical
distribution for moderate- $SO_2$ explosive eruptions in the lowermost stratosphere due to limited vertical sensitivity of
current satellite sensors (+/- 2 km accuracy) and low horizontal resolution of lidar observations. We found that using
revisited $SO_2$ injection heights and reduced $SO_2$-aerosol conversion timescale in a simple volcanic aerosol model
(EVA_H) improves SAOD estimates relative to available observations from the GloSSAC dataset. The protocol for
fast estimation of aerosol optical properties, radiative forcing and surface temperature response to volcanic eruption
has since been implemented in a seamless webtool (Volc2Clim, https://volc2clim.bgs.ac.uk/). The computationally
inexpensive nature of the webtool makes it ideal for rapid assessment of the volcanic climate effect and for
propagating large uncertainties that characterize early observations of volcanic clouds. Further development of the
underlying simple models as well as continued use of complex models explicitly modelling aerosol chemistry,
microphysics and transport remain critical given the complex nature of volcanic events. For example, the Raikoke
eruption took place in connection with two eruptions of Ulawun in June and August 2019 and just after a PyroCb




event which transported smoke into the stratosphere which were not considered in our original or revised

calculations. In addition, the recent HTHH eruption demonstrated that water vapor can also be injected into the

stratosphere which can affect $SO_2$ and aerosol lifetime but also with a radiative forcing that is opposite to volcanic

sulfate aerosols.

## Competing interests

The contact author has declared that none of the authors has any competing interests.

## Acknowledgement.

JPV and HC were supported by the NASA Roses program through the SAGE III Science Team (80NSSC21K1195)

and Upper Atmosphere Composition Observations program (80NSSC21K1082). TJA was supported by a global

mobility grant from the University of Exeter and a travel award from the Canada-UK foundation. ATP

acknowledges funding from the Natural Environment Research Council (NERC) R4Ash project (NE/S003843/1).

The Volc2Clim tool was kindly supported by the UK Earth System Modelling project, funded by the UKRI –

Natural Environment Research Council (NERC) national capability grant number NE/N017951/1 and the Met

Office, as well as NERC grants NE/S000887/1 (VOL-CLIM) and NE/S00436X/1 (V-PLUS). The GloSSAC data

were obtained from the NASA Langley Research Center Atmospheric Sciences Data Center.  The Volc2Clim

webtool is available at https://volc2clim.bgs.ac.uk/, and the source code is available on GitHub at

https://github.com/cemac/volc2clim/. The source code of the EVA_H volcanic aerosol model is available on GitHub

at https://github.com/thomasaubry/EVA_H. The source code of the FaIR climate model is available on Github at

https://github.com/OMS-NetZero/FAIR.

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



**Figures.**

751

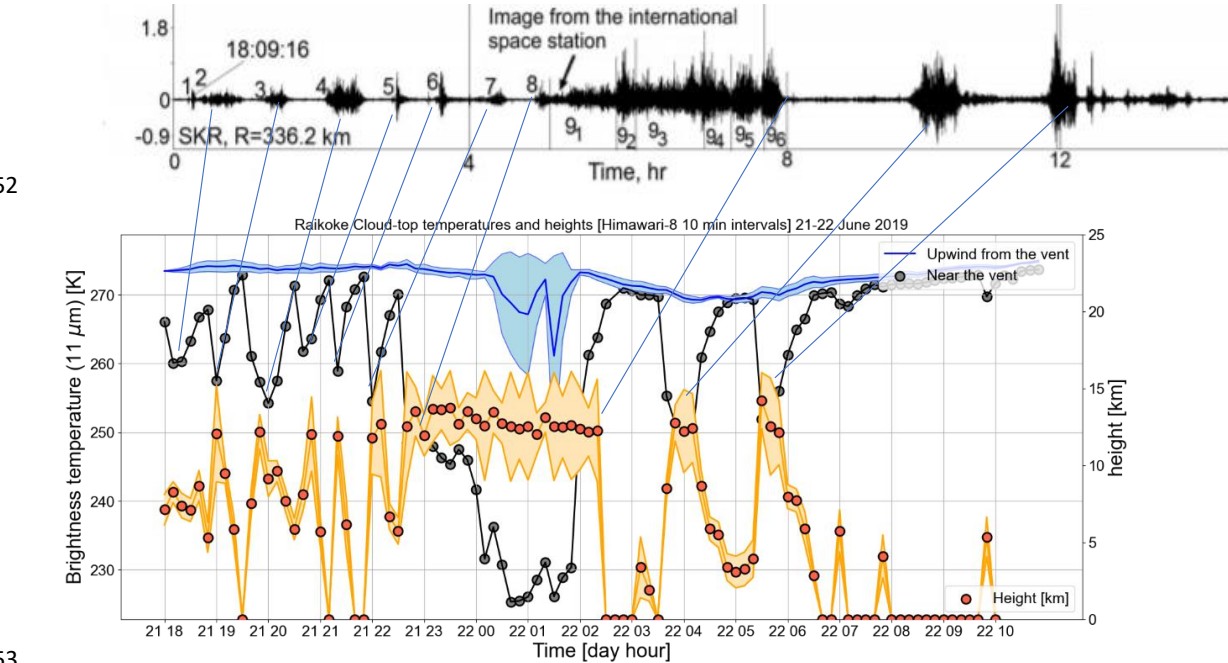

753

**Figure 1. (Top) Modified from Fig.7 from (Firstov et al., 2020) showing IS signals during the first 12h after the beginning of the Raikoke eruption which started near 18 UTC on June 21 2019. (Bottom) A time series of corresponding Brightness Cloud Top Temperature at 11μm derived from HIMWARI-8 is shown. Height retrievals near the vent (orange data points) and uncertainties (orange shaded region) taken from Prata et al. (2022).**













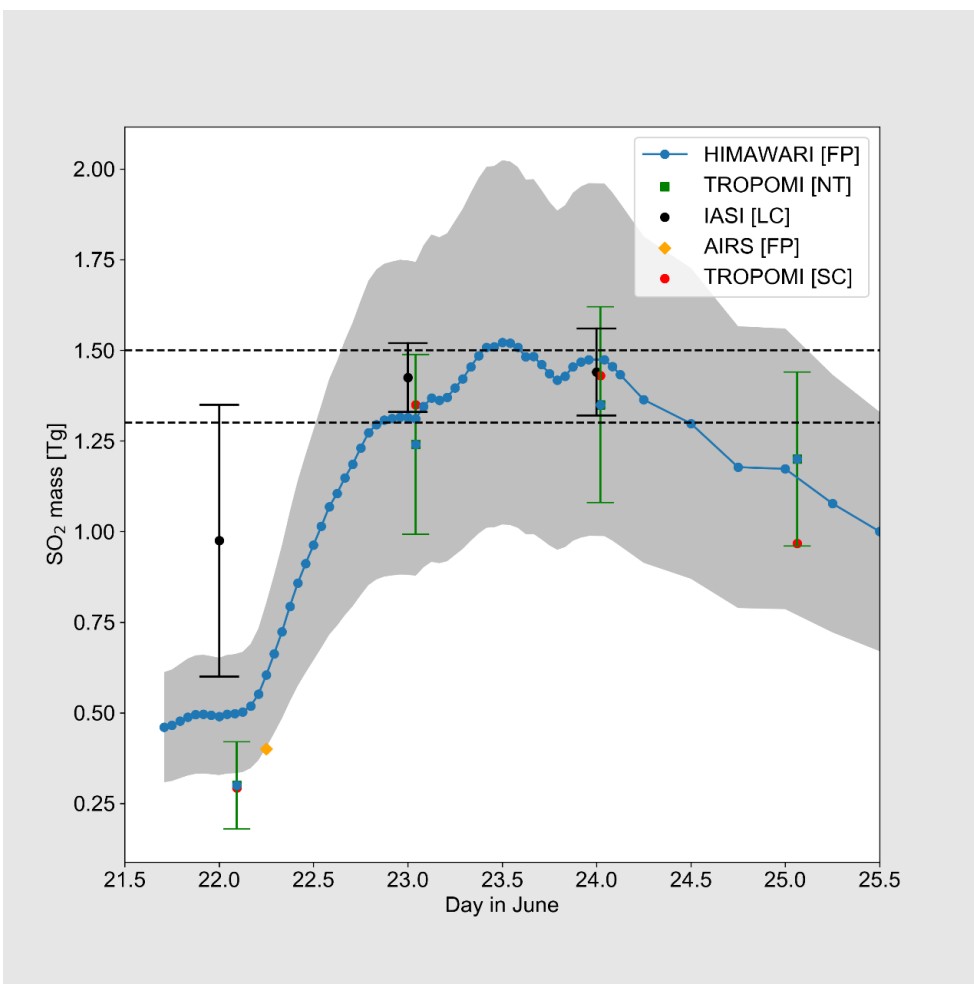


**Figure 2. Total $SO_2$ mass (Tg) as a function of time in June 2019 estimated from various satellite sensors for**

**the eruption of Raikoke. The grey-colored region indicates the uncertainty range of the Himawari-8 (AHI)**

**retrievals. A ±20% uncertainty has been placed on the TROPOMI estimates. The IASI estimates come from**

**different satellites and times of day (day/night); the vertical lines on these data indicate the range of the**

**estimations. Himawari-8 samples every 10 minutes. After 24 June retrievals were performed at longer**

**intervals. Distributed to VolRes on 06/28/2019.**






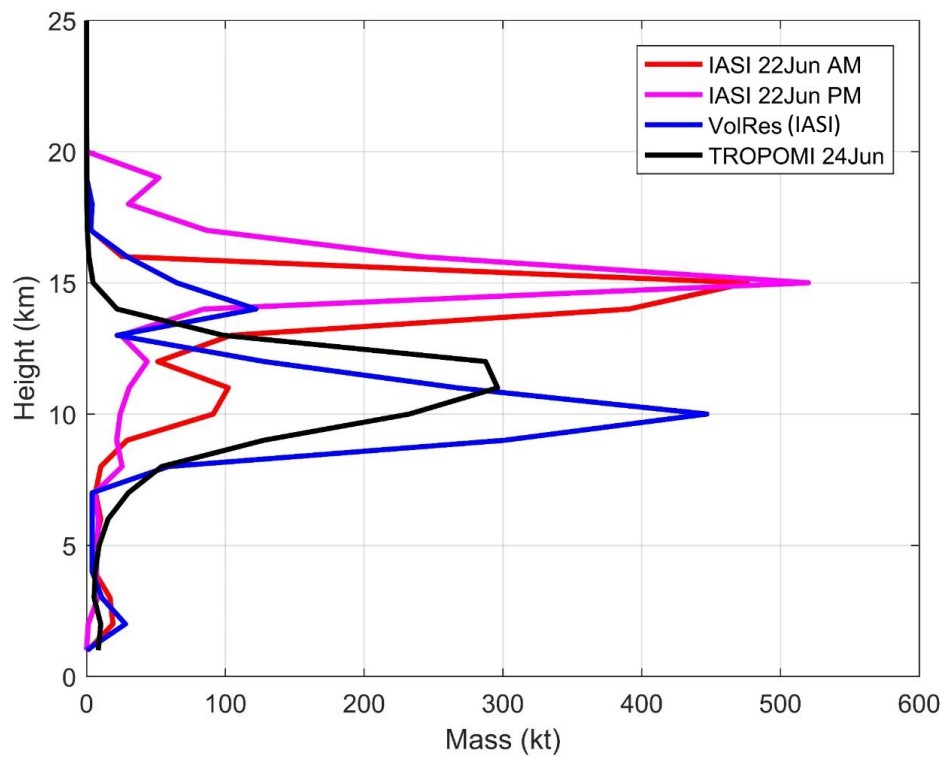


**Figure 3:** SO$_2$ **mass altitude distribution from IASI (refined analysis), VolRes (IASI initial estimate) and**

**TROPOMI. The associated data is provided in Table 1.**














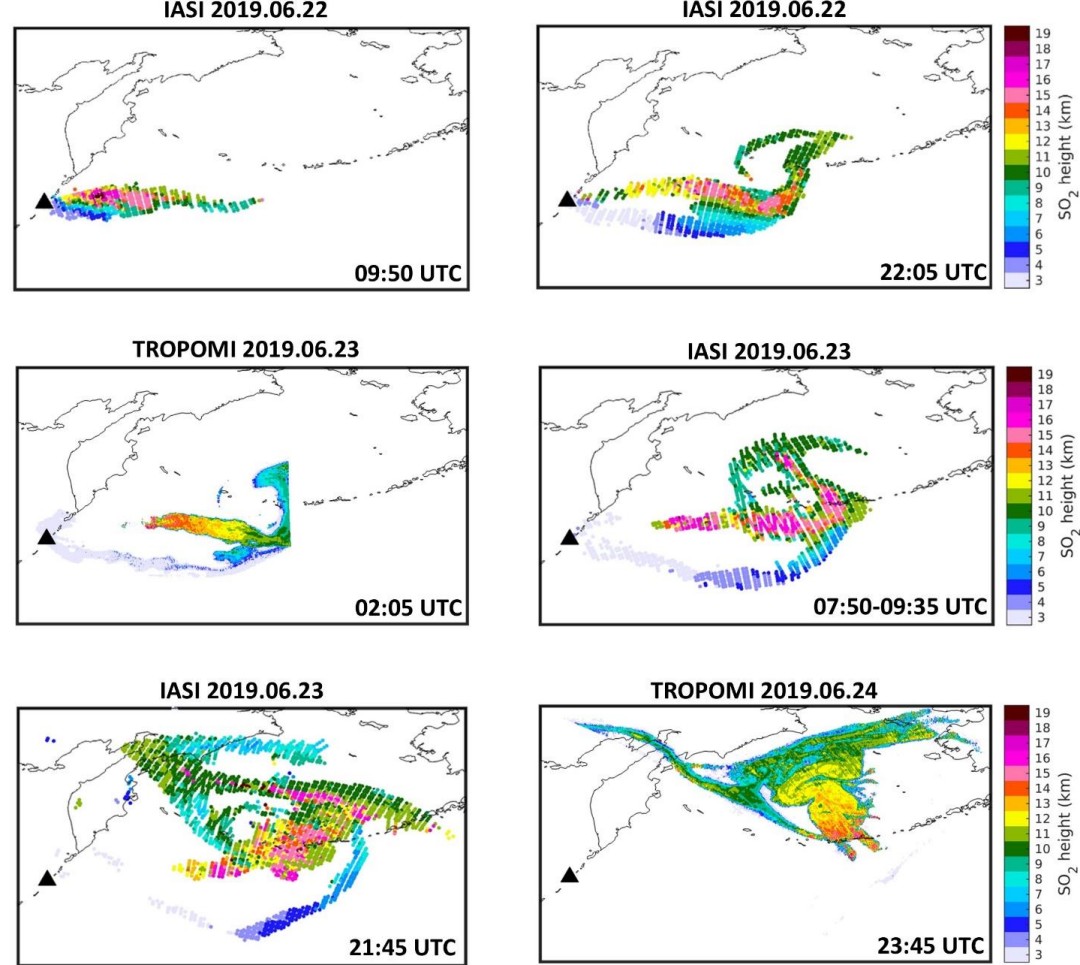


**Figure 4: Examples of** $SO_2$ **height retrievals from IASI (refined analysis) and TROPOMI for Raikoke eruption for 22-24 June 2019. The Raikoke volcano is marked by a black triangle. Approximate overpass times are indicated in each panel.**









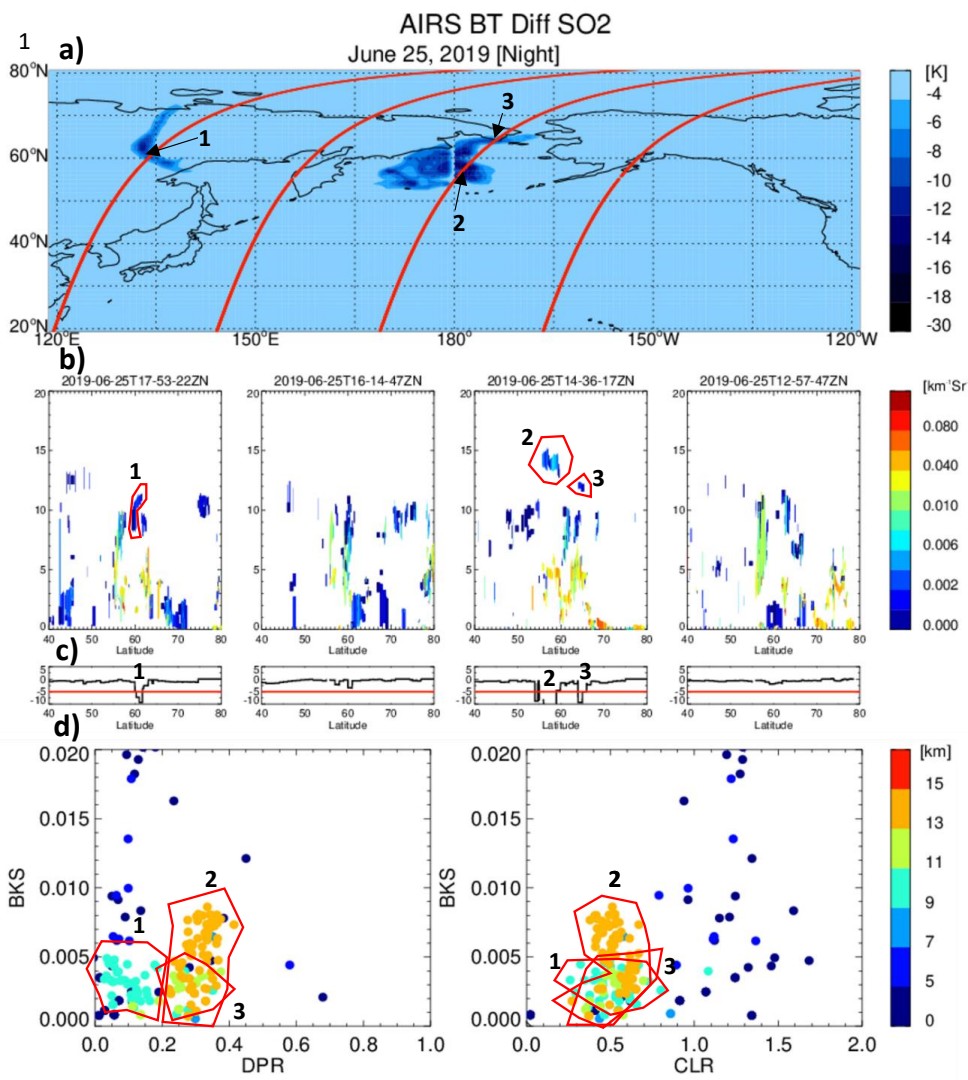

**Figure 5. (a) AIRS Nighttime Brightness Temperature Difference (BTD) (1361.44-1433.06 cm⁻¹) on 25 June 2022 together with 4 CALIOP ground-tracks (red). (b) Corresponding aerosol and cloud layer products from CALIOP level 2V4.2 product and (c) extracted AIRS BTD extracted along the CALIOP orbit tracks. (d) diagrams of particular backscatter (BKS) as a function of mean layer particulate DePolarization Ratio (DPR) (left) and particulate CoLor Ratio (CLR) (right) derived from CALIOP and colored by mid-layer altitudes.**



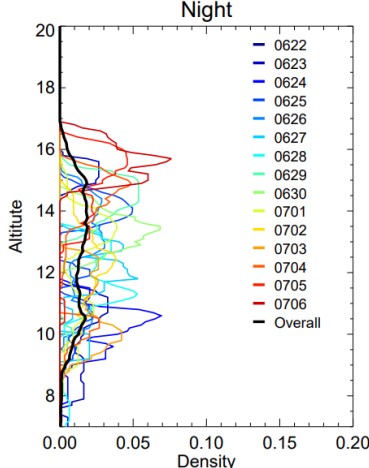

813

**Figure 6. Daily nighttime Probability Density Function profiles of the mid-layer geometric altitude for volcanic layers observed by CALIOP/AIRS using plume identification criterion when DPR < 0.4 and CLR < 0.7 and altitude > 5km and BTD < -6K between 06/22 and 07/06. The black line is the overall pdf profile using all nighttime data between 06/22 and 07/06.**







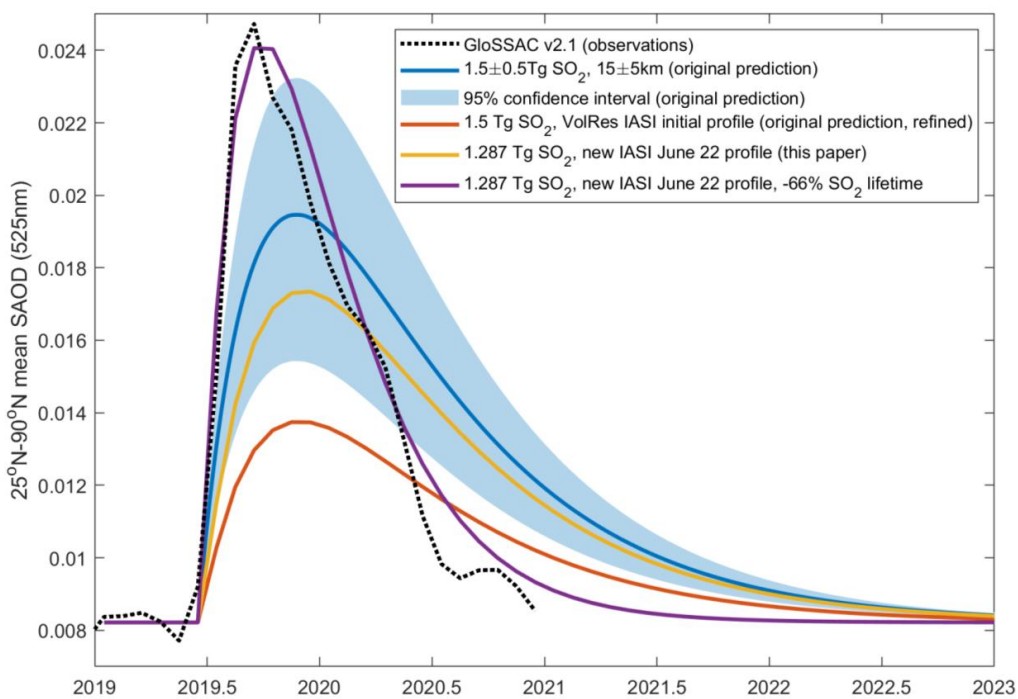


**Figure 7: Northern Hemisphere (25ºN-90ºN) monthly-mean SAOD at 525nm as projected by EVA_H (continuous colored lines) and observed (GloSSAC v2.1, black dashed line). The light blue shading and line shows the first projection made at the time of the eruption and its confidence interval based on an injection height of 15+/-5km and $SO_2$ mass of 1.5+/-0.5 Tg. The yellow line shows the second projection made at the time of the eruption using the VolRes IASI initial profile. The orange line shows a new projection using the new VolRes IASI June 22 profile presented in this study (Figure 3). The violet line uses the same profile, but the $SO_2$-to-aerosol conversion timescale in EVA_H reduced by 66%.**






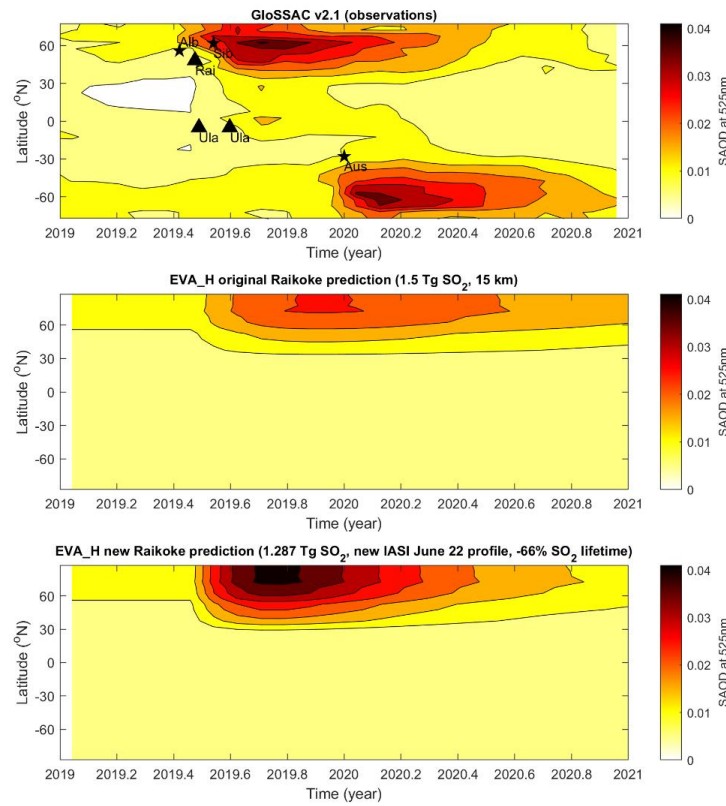

**Figure 8: SAOD at 525nm as observed (GloSSAC v2.1, top) and projected by EVA_H following the Raikoke 2019 eruption (middle) and using the revised IASI June 22** $SO_2$ **profile presented in this paper along with the adjusted (-66%)** $SO_2$**-to-aerosol conversion timescale in EVA_H (bottom). EVA_H was run only with the Raikoke injections, and not with injections associated with the Ulawun 2019 eruptions (denoted by black triangles in the top panel) nor with wildfire events in Alberta, Siberia (2019) and Australia (2020) (denoted by black stars in the top panel).**




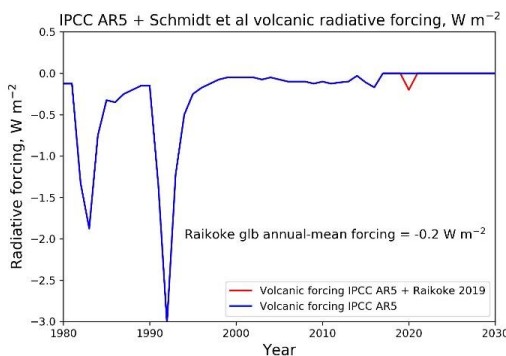
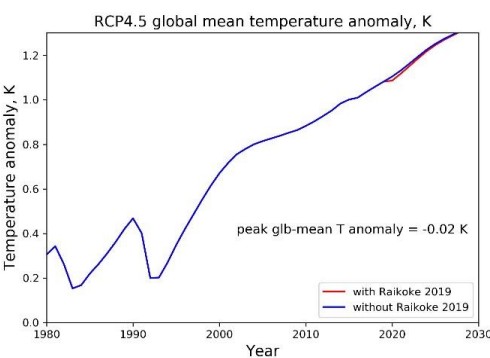

**Figure 9: Annual global mean volcanic radiative forcing (left) and corresponding annual global mean surface temperature anomaly calculated using the climate response model FaIR (Smith et al., 2018) (right). Blue and red lines show results with and without accounting for the 2019 Raikoke eruption, respectively. This is the original figure shared on the VolRes mailing list on 06/26/19.**






| Altitude | VolRes IASI initial profile | IASI 22 June 2019 (AM) | IASI 22 June 2019 (PM) | TROPOMI 24 June 2019 |
|---|---|---|---|---|
| 1 | 0 | 1.1 | 0 | 8.4 |
| 2 | 28 | 19.0 | 1.2 | 10.2 |
| 3 | 11 | 16.9 | 8 | 5.4 |
| 4 | 4 | 5.6 | 7.1 | 6.3 |
| 5 | 4 | 6.0 | 7.9 | 9.0 |
| 6 | 4 | 10.2 | 8.5 | 15.5 |
| 7 | 4 | 6.4 | 6.0 | 30.1 |
| 8 | 59 | 10.3 | 25.6 | 54.1 |
| 9 | 301 | 29.2 | 21.7 | 127.6 |
| 10 | 446 | 91.3 | 24.2 | 232.6 |
| 11 | 266 | 102.1 | 30.7 | 296.2 |
| 12 | 128 | 51.3 | 43.7 | 287.5 |
| 13 | 22 | 104.4 | 24.8 | 98.4 |
| 14 | 122 | 390.9 | 84.5 | 22.0 |
| 15 | 65 | 476.2 | 520.2 | 4.7 |
| 16 | 29 | 25.5 | 239.7 | 1.63 |
| 17 | 3 | 3.3 | 86.4 | 0.53 |
| 18 | 4 | 2.6 | 30.2 | 0.19 |
| 19 | 0 | 0 | 52.1 | 0.14 |
| 20 | 0 | 0 | 0 | 0.1 |
| **Total** | **1500 kt (scaled)** | **1352.3 kt** | **1222.5 kt** | **1210.6 kt** |


**Table 1:** $SO_2$ **mass profile (in kt) derived from IASI and TROPOMI for the Raikoke eruption.**



| Date | Data type | Activities | Data variables | Platform | Add. Information |
|------|-----------|-----------|----------------|----------|-----------------|
| 06/24 | Satellite | SO2 and plume height maps 06/24 & 06/25 | SO2 total column (DU) and concentration (ppmv ?) | TROPOMI /Sentinel 5P | Polar Orbit/ESA |
| 06/24 | Satellite | Aerosol maps and profiles when ? | Aerosol extinction (km-1) | NPP/OMPS | Polar Orbit/NASA |
| 06/25 | Satellite | SO2 maps 06/21 & 06/22 | SO2 total column (DU) | Metop/IASI | Polar Orbit/Eumetsat |
| 06/25 | Satellite | Ash and SO2 total column | Ash signature (11-12 um) and SO2 UTLS (VCD DU) | AHI/HIMAWARI-8 | Geo Orbit/JAXA |
| 06/25 | Satellite | Plume heights and optical properties | Backscatter and depolarization at 532 and 1064 nm | CALIOP/CALIPSO | Polar Orbit/NASA |
| 06/25 | Satellite | Maps of plume height and properties 06/23 | Height (km) and AOD, angstrom coeff, SSA | MISR/Terra | Polar Orbit/NASA |
| 06/25 | Model | Volcanic plume maps at 100 and 140 hPa | Aerosol extinction at XX nm | WACCM | Model type |
| 06/25 | Model | Impacts on stratospheric aerosol | Stratospheric AOD | GEOS-5 | |
| 06/26 | Satellite | Mass distribution profile on 06/23 | Mass per levels (kt) | TROPOMI/Sentinel 5P | Polar Orbit/ESA |
| 06/26 | Satellite | SO2 plume vertical information | SO2 mixing ratio (ppbv) | MLS/Aura | Polar Orbit/ESA |
| 06/26 | Model | Radiative and climate impacts | RF TOA (w/m2) | ?? | |
| 06/28 | Model | Trajectory simulation of Raikoke dispersion | Plume height (km) | Langley Trajectory Model | GEOS-5 wind data |
| 07/03 | Satellite | Plume height and properties | Backscatter and depolarization at 532 and 1064 nm | CALIOP/CALIPSO | Polar Orbit/ESA |
| 07/09 | Model | SO2 and ash plume dispersion 06/21 to 06/25 | Ash and SO2 mass concentration | ICONN-ART | |
| 07/10 | Ground-based lidar | Vertical plume profiles 07/05 | Scattering ratio at 532 nm | OHP/LTA | |
| 07/10 | Satellite | Plume height and properties | Backscatter and depolarization at 532 and 1064 nm | CALIOP/CALIPSO | Polar Orbit/NASA |
| 07/10 | Satellite | Latitudinal time series | Aerosol extinction (km-1) | NPP/OMPS | NASA |
| 07/16 | Satellite | Animation of aerosol maps at 12.5 km, 13.5 km, 14.5 km and 16.5 km across the NH. 06/11 to 07/14 | Aerosol extinction (km-1) | OMPS/NPP | Polar Orbit/NASA |
| 07/17 | Ground-based lidar | Volcanic aerosol profiles 06/29 and 07/08 | RSC 1064 nm | SIRTA | |
| 07/19 | Satellite | Maps of SO2 centered in Indonesia/Australia (from 06/26 to 07/12), Ulawun eruption | SO2 DU | TROPOMI /Sentinel 5P | Polar Orbit/ESA |
| 07/20 | Satellite | Animation of aerosol maps at 18.5 km from 06/27 to 07/17 | Aerosol extinction (km-1) at 674 nm | OMPS/NPP | Polar Orbit/NASA |
| 07/21 | Ground-based lidar | Volcanic aerosol profiles on 07/18 and 07/20 | Scattering Ratio at 532 nm | OHP LTA | |
| 08/07 | Satellite | Animation of aerosol maps at 20.5 km | Aerosol extinction (km-1) at 674 nm | OMPS/NPP | Polar Orbit/NASA |
| 08/24 | Satellite | Volcanic plumes cross-section 11-20 Aug 2019 | Scattering Ratio at 532 nm | CALIOP/CALIPSO | Polar Orbit/NASA |
| 09/04 | Balloon | Aerosol concentration profiles on 08/26 in Wyoming | Aerosol concentration for r>0.005 um, 0.092, 0.15, 0.28 | Balloon | WOPC |
| 09/17 | Ground- | Atmospheric profiles of aerosols and | Backscatter profiles at 532 nm | Lidar LOA | |



**Table 2: VolRes activities during the first 2 months after the Raikoke eruption.**