# Peer review of "The 2019 Raikoke eruption as a testbed for rapid assessment of volcanic atmospheric impacts by the Volcano Response group"

_EGUsphere, 2023_

## Community Comment (CC1)

**Review: Vernier et al., egusphere-2023-1116, 2023**

The manuscript mainly deals with the perturbation of the stratospheric aerosol layer after the volcanic eruption of the Raikoke volcano in June 2019. The authors assume that the perturbation was (almost) exclusively caused by the conversion of emitted SO2 into sulfate aerosol.

The main results are presented in Figures 7 and 8. Figure 8 shows that the main perturbation occurred between 45°N to 90°N.

At the same time (July to December 2019), strong Siberian fires contributed to the UTLS aerosol load over the Arctic (at least from 60°-90°N), as reported by Ohneiser et al. (2021) and Ohneiser et al. (2023) for the latitudes from 85° to 90°N and by Ansmann et al. (2023) for the latitudes from 60°-90°N. See also Figure 1 in the review comment. Figure 1 is taken from Ansmann et al. (2023).

Part of the SAOD in Figure 8 was probably caused by Siberian smoke. The authors did not consider this additional pollution aspect in their study. According to Figure 1 in this review, the smoke contributed about 75% to the UTLS aerosol (mostly below 12 km height) and the Raikoke aerosol around 25% in the autumn of 2019 (mostly above 12 km height).

The presence of smoke had obviously an impact of the measured decay times in Figure 7 of the manuscript. The decay of the stratospheric perturbation after the Raikoke eruption should be similar to the decay after the rather similar Sarychev volcanic eruption in 2009. For the neighbor volcano Sarychev, that erupted exactly 10 years before Raikoke, Haywood et al. (2010) determined a decay time of 90-150 days (3-5 months). Now the decay time is 12 months or even longer according to Figure 7. The major volcanic Pinatubo eruption caused decay times of about 12-15 months because the sulfate plumes reached heights of 25-30 km, and sedimentation and removal then needs longer as for sulfate layers with top heights around 20-22 km (Raikoke aerosol).

Is it not possible that the presence of smoke prolonged the occurrence of pollution in the UTLS height range? Ohneiser et al. (2021) reported smoke until May 2020 over the central Arctic.

The following question needs to be discussed: Why is the decay time now 12 months and more and thus much longer than the 3-5 months in the case of the Sarychev sulfate aerosol?

The following aspect needs to be included in the discussion: The strong Siberian fires in the summer of 2019 and the resulting record-breaking smoke pollution of the Arctic troposphere and lower stratosphere (at least north of 60°N) in the second half of 2019 needs to be mentioned and the potential consequences for the simulations need to be discussed. In Ohneiser et al. (2023), several references are given that deal with the strong smoke burden over Arctic latitudes in 2019.

[Figure]

Figure 1: The Arctic UTLS aerosol layer in the fall of 2019 (Ansmann et al., 2023). The black solid line shows the 532 nm particle extinction coefficient (October-November mean profile) measured with ground-based Raman lidar during the MOSAiC expedition at 85°-86°N (Ohneiser et al., 2021). The blue and green profiles show October 2019 mean extinction profiles for the latitudinal bands from 60°-70°N (green) and 70°-85°N (blue) measured with the satellite-based ACE instrument (1020 atmospheric transmission channel) (Boone et al. 2022). The 1020 nm extinction profiles are multiplied by a factor 3. We hypothesize that Raikoke sulfate aerosol dominated at stratospheric heights above 12 km (dotted line) and wildfire smoke dominated in the upper troposphere up to the extinction maximum. AOTs for 532 nm measured at 85-86°N are given as numbers.

Ohneiser, K., et al., The unexpected smoke layer in the High Arctic winter stratosphere during MOSAiC 2019–2020, Atmos. Chem. Phys., 21, 15783–15808. doi: 10.5194/acp-21-15783-2021, 2021.

Ohneiser, K., et al., Self-lofting of wildfire smoke in the troposphere and stratosphere: simulations and space lidar observations, Atmos. Chem. Phys., 23, 2901–2925, doi: 10.5194/acp-23-2901-2023, 2023

Boone, C. D., et al., Stratospheric Aerosol Composition Observed by the Atmospheric Chemistry Experiment Following the 2019 Raikoke Eruption, JGR-Atmospheres, 127, doi: 10.1029/2022JD036600, 2022.

Ansmann, A., et al., Comment on ``Stratospheric Aerosol Composition Observed by the Atmospheric Chemistry Experiment Following the 2019 Raikoke Eruption'' by Boone et al., JGR-Atmospheres, 128, 2023, and will be published in August or September 2023.

---

## Community Comment (CC2)

I am reluctant to wade in here but feel the need to clarify an issue under discussion. I am in the process of submitting a reply to Dr. Ansmann's comment on my JGR paper describing Raikoke sulfate aerosols (referenced in his comment to the Vernier et al. paper), where I will make the case that aerosols in the Arctic following the Raikoke eruption are predominately sulfates and not smoke. I would advise people to wait for that reply before drawing any final conclusions on the subject. The Ansmann comment on the Vernier et al. paper leans heavily on a perceived significant discrepancy with the evolution of sulfate aerosols from the Sarychev eruption, but if one looks at measurements of both events, there is no such discrepancy.

Both Raikoke and Sarychev are located around latitude 48 °N. Raikoke erupted June 22$^{nd}$, 2019, while Sarychev erupted June 11-21, 2009, the same time of year. One would expect the evolution of sulfate aerosols following the eruption to be quite similar for the two events. Figure 1 shows the monthly average ACE-Imager 1.02 μm extinction profiles (which can be used as a proxy for aerosol loading) for the latitude range 60-75 °N, looking at the variation over time for the periods following the two eruptions. In each panel, a profile from a different year is included to give a sense of what the extinction would be in the absence of elevated aerosols. The extinction from Raikoke is larger because it was a larger eruption, but changes in aerosol loading as a function of time for the two events track each other very well. The Ansmann comment gives the impression that Sarychev sulfate aerosols are completely gone in a few months, but they remain elevated for close to a year after the eruption, just like Raikoke.

As a means of independent verification, Figure 2 shows average extinction plots for 750 nm for the latitude range 77-83 °N from the OSIRIS instrument on Odin, another instrument that measured aerosols following both eruptions (https://research-groups.usask.ca/osiris/data-products.php). The results are in generally good agreement with the ACE observations. OSIRIS has no measurements at high northern latitudes between October and March, but there remains a clear enhancement in aerosol levels in March 2010, 9 months after the Sarychev eruption.

According to the arguments presented by Dr. Ansmann in the comment to my JGR paper on Raikoke, extinction by Sarychev sulfate aerosols should be more than a factor of 4 smaller than that for Raikoke by October. That effect is not evident in either the ACE or the OSIRIS measurements.

[Figure]

**Figure 1:** Average ACE-Imager 1.02 µm aerosol extinction in the latitude region 60-75 °N for the periods following the Sarychev and Raikoke eruptions, along with a profile showing extinction where aerosols were closer to background levels.  a) July.  b) September.  c) October.  d) November.  e) January.  f) February.  Note the different horizontal scales.

[Figure]

**Figure 2:** Average monthly OSIRIS 750 nm aerosol extinction for the latitude range 77-83 °N for the periods following the Sarychev and Raikoke eruptions, along with a profile to give a sense of background levels. Note the changing x-axis scale.

---

## Author Comment (AC1)

Response to Dr. Ansmann and Dr. Boone.

First, we would like to thank them for their interest in our paper and finding time to read and comment on it. This is very much appreciated. Given the fact that the comment from Dr. Boone is a response to Dr. Ansmann, we will provide an answer to both comments in the same document.

Dr. Ansmann raised an interesting point regarding the decay time of the Raikoke plume compared to the Sarychev eruption. He found that the decay time of Raikoke was significantly longer (12 months) than the one observed after the Sarychev eruption (3-5 months) as derived by Haywood et al. (2010). However, Dr. Boone provides compelling evidence in his reply that the extinction profiles at 1.02 µm from ACE Imager behave in a similar way for both eruptions so that Sarychev decay is not much faster than Raikoke. As a result, it's not required to invoke additional aerosol sources to explain the decay time of Raikoke.

However, Ohneiser et al. (2021) show Raman lidar profiles (Fig. 11/12) indicating that the optical properties of the aerosol layers observed during the MOSAiC project were consistent with smoke layers. Indeed, the lidar ratio at 532 nm is expected to be close to 50sr, a value usually assumed for the sulfate aerosol background conditions to convert backscatter coefficient into extinction. Mattis et al. (2010) found a value similar (40 sr) using ground-based lidar over Germany after several volcanic eruptions. However, Ohneiser found a lidar ratio at 532 nm near 80 sr and thus apparently significantly higher than expected. This would be an indication of the presence of smoke. While Ohneiser al. (2021) still claims that the Raikoke plume is likely located above 12-13 km at the top of the smoke layer, they do not report optical properties to confirm that leaving this very important aspect unclear. In addition, one would expect some sort of transition between two layers displaying different optical properties especially several months following events due to microphysical processes (coagulation, growth, sedimentation). Instead, profiles provided in Ohneiser et al. (2023) and in Drs. Boon/Ansmann's comments show a good continuity along the vertical profile.

In addition, our team conducted several balloon flights from Hampton, VA in October and November 2019 after the Raikoke eruption (paper not yet published). We included profiles of aerosol concentrations for several size bins. While Hampton (37.09, -76.37) is far from the locations of the MOSAIC project located in the Artic region, we could expect that that residual smoke from the summertime in Siberia could have been transported across the Northern Hemisphere after several months. The concentration profiles shown below indicate an aerosol layer between 15-24 km (above the local tropopause ~12km). We note that this layer does not display any significant changes in the vertical structure of the size channels where most of the aerosols size range within 0.15-0.25 µm. The thickness of the layer and its position relative to the tropopause is consistent with the one reported in the comments by Dr. Ansmann and Dr. Boone as well as in Ohneiser et al. (2021) but with a peak at higher altitude given the isentropic transport expected from polar region to mid-latitudes.

[Figure]

Particle Concentration profiles for aerosol radius greater than 0.15, 0.25, 0.5, 1.25, 2.5 and 5 μm obtained in Hampton VA on 4 October 2019 using the Particle plus Optical Particle Counter (see description in Dumelie et al., 2023, Li et al., 2023).

In summary, we added in the manuscript a comment mentioning Ohneiser et al. (2021) and the potential role of smoke on aerosols near the tropopause after the Raikoke eruption but believe that a major and prolonged influence of smoke is unlikely and that most of the SAOD is likely from the 2019 Raikoke. Nevertheless, we cannot rule out an influence from wildfire Siberia smoke.

**References**

Dumelié, N., and Coauthors, 2023: Toward Rapid balloon Experiments for sudden Aerosol injection in the Stratosphere (REAS) by volcanic eruptions and wildfires. *Bull. Amer. Meteor. Soc.*, https://doi.org/10.1175/BAMS-D-22-0086.1, in press.

Li, Y., Pedersen, C., Dykema, J., Vernier, J.-P., Vattioni, S., Pandit, A. K., Stenke, A., Asher, E., Thornberry, T., Todt, M. A., Bui, T. P., Dean-Day, J., and Keutsch, F. N.: *In situ* measurements of perturbations to stratospheric aerosol and modeled ozone and radiative impacts following the 2021 La Soufrière eruption, EGUsphere [preprint], https://doi.org/10.5194/egusphere-2023-1891, 2023.

---

## Author Comment (AC2)

Review of "The 2019 Raikoke eruption as a testbed for rapid assessment of volcanic atmospheric impacts by the Volcano Response group" by Vernier et al. for publication in EGUsphere.

The paper presents a sort of process/discussion event of the rapid analysis of the June 2019 Raikoke volcanic eruption through the perspective of the "Volcano Response" group, a grassroots community communicating via an email list and attempting to provide information quickly to the scientific community following volcanic events. Several datasets are synthesized toward providing initial and later, refined estimates of the volcanic plume vertical profile and loading of sulfur dioxide (SO2) and aerosol. Impact of initial and refined inputs for plume altitude and loading of SO2 (and so resulting aerosol) are presented in Figure 7, and the impacts on temperature and radiative forcing are presented Figure 9. Overall it is argued that the Raikoke eruption has small radiative forcing impact and so small impact on surface temperature.

As a process the paper is of sufficient interest to warrant publication after minor revisions. The paper does need a pretty thorough reading for copy editing. I note below a number of places where clarity of the text can be improved and further clarifying information should be provided instead of leaning so heavily on community jargon. Some of the figures are lacking in legends or labeling so as to not be intuitive what is being shown.

We would like to thank reviewer 1 for his comments for which responses are provided below.

4: affiliation #3 arrives after #4 in the ordering. Please revise.

Corrected

56: "hellmouth," — add comma

Corrected

60: "(SVERT)," — add comma

Corrected

64: "red warnings for aviation" — what does this mean, please explain

**The following text was added to the paper to further explain the role of KVERT**

"In addition, KVERT which issue volcano observatory notice warning for aviation had flagged with an aviation color code red which signifies that an "eruption was underway with significant emission ash into the atmosphere" (see KVERT webpage for more information http://www.kscnet.ru/ivs/kvert/van/index?type=1)"

67: Figure 1 needs some work to be understandable. There are no axis labels or units on the top panel, and neither does the text or caption explain what "Infrasound Signal" means. Further, eleven explosive episodes are mentioned in the text, but only the first 9 are labeled in the panel. I have no idea how to interpret the bottom panel. Is the blue line the cloud-top temperature and goes with the lefthand axis? What do the orange and grey dots represent? This needs a much clearer legend and further explanatory text in the caption at least.

Figure 1 has been improved to respond to these comments and addition text in caption is provided.

69: Please explain more clearly what "(1,2,3,7,9 and 10)" refers to. I think these are the explosive events. #10 is not labeled in Figure 1a (although I can count, so I suppose…) but I'm still not sure if it is meant that this group of episodes is "vent outflow" or the other kind.

Yes, 1,2,3 refers to explosive events. Indeed #10 and #11 have been added the figure caption was modified as well as the text.

83: "(SSIRC)," — add comma

added

89: This is a different URL than in line 104, though they go to the same place. Suggest you state same URL for less confusion. Also, as written here there is an extraneous superscript "2" at the end of the URL that should be removed. And I note the page was initially confusing as it does not appear to show any meaningful content on mobile browser, but does on desktop.

This was fixed.

105: I note here that you refer to Table 2 before you ever refer to Table 1 (line 196). Suggest you reorder the tables accordingly.

Yes, tables have been reordered.

120: "Precursor satellite," — add comma

added

120: "on board" — line 127 writes "onboard" while 140 and 148 both write "on board" as here. Be consistent.

This is been corrected by using "onboard"

141: "equatorial" — do not capitalize

Corrected.

151: "Cloud layer products," — add comma

Added.

152: CAD "less than -100 or greater than -20" — expand on what this means or otherwise clarify. This is not meaningful to the average reader.

We expanded about the CAD in the text and provided additional reference

154: "0, 1, 16, and 18 are rejected" — same comment as above, what does this mean?

Additional information is provided in the main text.

157 - 171: This is a repeat of the text in 66 - 80 and should be removed.

This was removed.

173: I note here you skip to Section 4 and there is no Section 3. Subsequent numbers follow Section 4.

This is now fixed.

183: "as will be discussed" — add "be"

Fixed.

185: Suggest start a new paragraph at "Figure 2 shows…"

Changed.

188: "temporal evolution than the one" — replace "than" with "as"

Replaced.

227: What does "bulk height" mean? Center of mass?

This refers to the mean position of the plume height in terms of mass.

So it was changed by "center of mass" for clarity.

232: "effective heights" — again, what does this mean? Here and previous comment please establish a definition and be consistent so that data are not mis-applied.

Regarding the wording 'effective height', it comes from the fact that the retrieved height is the result of a minimization process between simulated and measured radiances. Any $SO_2$ height algorithm considers a simplified model of the radiative transfer in the atmosphere. Therefore, it is unable to represent fully the observation condition and complex photon path in a volcanic plume (multi-layered, mixture of aerosols of different types, sizes, etc). IR and UV have different sensitivity to different influence parameters so we should not expect that the retrievals give the same results

256: "eruption, allowing" — add comma

Added.

258: I would say "low spatial coverage" is more appropriate than "low horizontal resolution." It's a coverage matter rather than a resolution issue.

Yes, we agree with this comment and modified the text accordingly with the term "low spatial coverage".

269: Here and throughout paragraph, since you labeled Figure 5 in parts a, b, c, ... suggest you refer explicitly to those panels by letter so it is clear what to look at. Figure 5c I don't understand what the red line is, and there are no units given for the y-axis (I infer K).

An explanation is provided in the caption about the red line. We also added the unit for y-axis of figure 5c.

275: "second orbit" — not clear what you mean, as this is the third orbit from left to right. You describe the leftmost (latest) orbit first. Maybe the plume was encountered on "a second orbit"?

Labelled about orbit number has been added to the figure to clarify this and the text has been modified to reflect which orbit is referred to.

281: Suggest start a new paragraph at "We visually inspected…"

We started a new paragraph.

290: "aerosol loading," — add comma

Added.

312: "Ghassan Taha" and "Clarissa Lieven"

Apostrophe.

315: "times, randomly" — add comma

Added.

354: You do not appear to refer to or discuss the left hand panel in Figure 9. Omit.

We refer to the number in that panel in section 6.2-6.3 but had not linked it explicitly to the figure. This has been corrected.

434: "HTHH" is being used, but has not explicitly been connected to the Hunga Tonga Hunga Ha'apei eruption and should be noted.

Now explicitly connected.

768: In the legend for Figure 1 there are parenthetical notations ([FP], [NT], …) with no explanation. Please either remove or explain.

Now fixed

808: Should be "particulate" instead of "particular"

Fixed.

826: I believe the figure and legend are correct, but caption refers here to the yellow line when the orange line is meant, and vice versa on the following line. Please correct or clarify.

Yes, this is now corrected.

872: Label or add in caption that altitude is in km.

Now added.

Citation: https://doi.org/10.5194/egusphere-2023-1116-RC1

Review 2

I agree with the comments of the other reviewer overall and have little to add. I highly recommend that the manuscript be professionally copy-edited.

Citation: https://doi.org/10.5194/egusphere-2023-1116-RC2

Thanks for this comment. We are not planning to have a professional editor other than ACP before publishing this paper.

---

## Author Response (AR2)

Figure 1 is improved, but the labeling in the paragraph lines 69 - 84 does not accurately point to the parts of the figure. Line 74 - change Fig 1b to Fig 1(top) and line 75 change Figure 1a to Figure 1 (bottom). You also now label a 12th explosive event in Figure 1 (bottom) but refer in the text only to eleven events.

Line 158: You just defined the CAD to be between -100 and 100, but now state you are excluding points with CAD less than -100. How does that work?

Regarding discussion of Figure 5, you could just refer to the orbit number in Figure 5a, that is now clear enough.

Line 320: I meant the names were presented oddly. "Ghassan Taha" instead of "Taha Ghassan", etc. I had a comment before that incorrectly referred to Figure 1 when I meant Figure 2. The legend in Figure 2 has some obscure notation in it. "Himawari [FP]" What is [FP]? I infer this refers to author initials, but it's weird and not relevant to include here.

Response

We acknowledge and appreciate reviewer 1 for his comments which we addressed below.

1) We modified Fig.1 and added panel a and b to align with the text. We also made a few corrections in the text about the 12 episodes.
2) The line about the CAD score has been clarified
3) Line 320 is now corrected
4) Figure 2 has been modified